# Amorphous Carbon Coatings with Different Metal and Nonmetal Dopants: Influence of Cathode Modification on Laser-Arc Evaporation and Film Deposition

**Tim Krülle** [1,2,*] , **Frank Kaulfuß** [1,*] , **Volker Weihnacht** [1], **Falko Hofmann** [1] **and Florian Kirsten** [1]

1    Fraunhofer Institute for Material and Beam Technology (IWS), 01277 Dresden, Germany; volker.weihnacht@iws.fraunhofer.de (V.W.); falko.hofmann@iws.fraunhofer.de (F.H.); florian.kirsten@iws.fraunhofer.de (F.K.)
2    Institute of Materials Science, Technische Universität Dresden, 01069 Dresden, Germany
*    Correspondence: tim.kruelle@iws.fraunhofer.de (T.K.); frank.kaulfuss@iws.fraunhofer.de (F.K.)

**Abstract:** In this study, the arc evaporation of pure graphite and composite cathodes with small amounts of metals (Mo, Fe) or nonmetals (B, Si) was investigated by means of a laser-arc process. Both specific aspects of the arc evaporation and the effects on the deposition of the doped and undoped carbon coatings were studied. The deposition rate, the chemical composition and the mechanical properties of the generated films were evaluated. In addition, the dependence of the deposition rate and the composition on the height position of the substrates in relation to the cathode were also the subject of the investigations. Finally, the erosion rate and the arc spot behavior on the cathode were analyzed. It is shown that homogeneously doped (t)a-C:X coatings can be reliably synthesized with the laser-arc technique. There are differences in the various properties of the coatings and the deposition rate. The latter is attributed in particular to the erosion behavior of the cathodes.

**Keywords:** laser-arc; tetrahedral amorphous carbon (ta-C); doped carbon coating; cathode erosion; metal-containing carbon; nonmetal-containing carbon; Diamond-like carbon (DLC)



## 1. Introduction

Tetrahedral amorphous carbon (ta-C) coatings have found a wide range of industrial applications, e.g., for tribological components, for cutting tools and in magnetic storage devices, due to the outstanding combination of several properties, such as high hardness, high Young's modulus, chemical inertness, good tribological behavior and special optical properties [1–3]. The deposition processes of carbon coatings are well understood, and comprehensive studies have been undertaken by Robertson or Schultrich [2,4]. Significantly, the cathodic arc evaporation of carbon is an appropriate process for the deposition of ta-C due to its high deposition rate, high ionization and the optimal energy range of the ions [5]. Limitations to the use of ta-C coatings are mainly due to the macroparticle emission [6] that occurs during the evaporation of graphite cathodes. To overcome this problem, filter systems [1] or post-treatment processes [7] can be utilized to minimize the effect of macroparticles and the resulting growth defects. An alternative way to minimize the macroparticle erosion itself is given by Kandah et al. [6,8,9]. In those works, the cathode material is optimized with regard to graphite grain size, the density of the graphite cathode and pore size (distribution), as well as the electrical resistivity of the graphite material. By optimizing these parameters, the arc spot velocity on the cathode surface is increased, leading to a reduction in the number of emitted macroparticles.

The doping of carbon coatings is a promising way to tailor coatings with regard to specific properties (see Figure 1). Several authors [2,10–13] investigate the effect of cathode modification and doping elements on the deposition and properties of carbon coatings, whereby the studies from Sánchez-López [11] and Zia [12] gave comprehensive

overviews. The effect of doping carbon coatings is described in order to improve the properties of the coatings with respect to intrinsic stress, crack resistance or temperature stability (for references, see Table 1). In most cases, the coatings are doped by modifying the graphite cathodes with other elements. Doping carbon films can be realized in various ways. Cathode modification through the infiltration of graphite cathodes with saline solutions or liquid metal containing the modifying element [14] is one option. This requires an open porosity capable of infiltration in order to store the liquid. Another option is using a mixture of graphite powder and a modifying element containing powder, which is pressed and sintered [15]. The selective variation of the content of dopants in the coating during the deposition is possible via the co-evaporation of carbon and dopant-containing target(s). Finally, carbon evaporation can be carried out in a gas atmosphere (with hydrogen-containing [2], nitrogen-containing [16] and silicon-containing gas [17], etc.). Common physical vapor deposition (PVD) processes, such as arc PVD or sputtering, as well as combinations, such as plasma-enhanced chemical vapor deposition (PECVD) processes, can also be used as deposition processes [1].

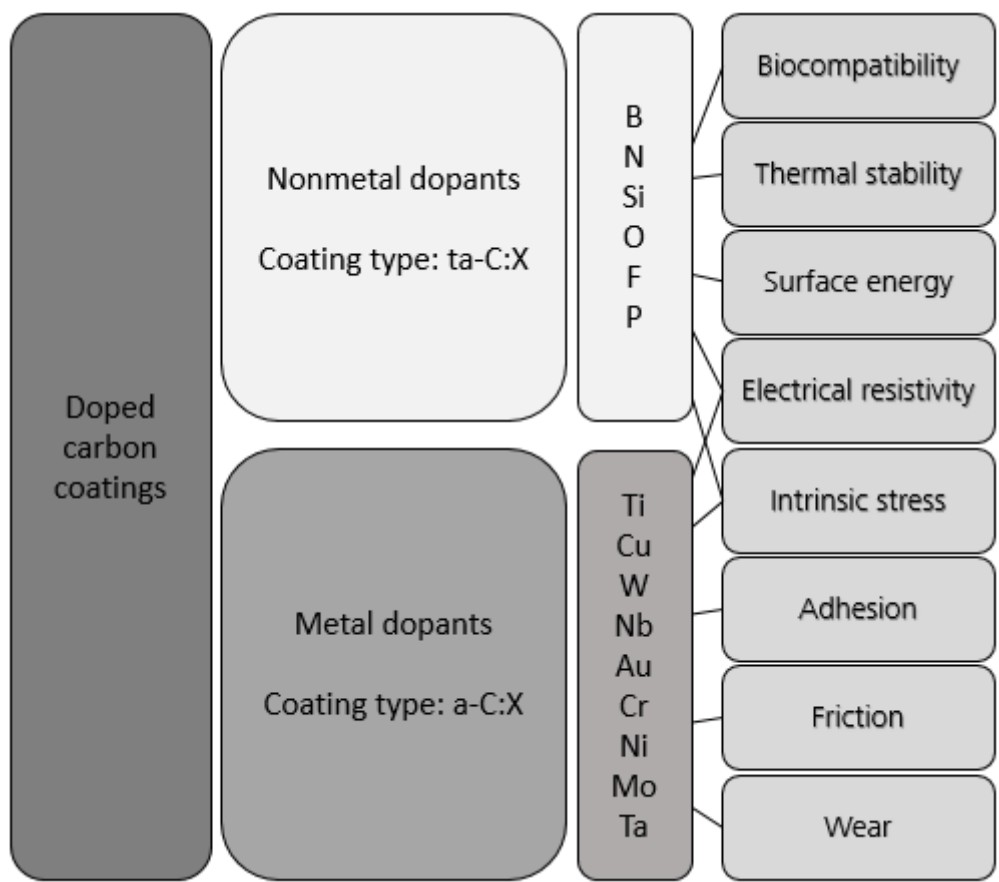

**Figure 1.** Effect of dopants on the properties of doped carbon coating; modified from [11].

Metal doping often results in a softening of the carbon matrix, shifting the bonding state from $sp^3$ to $sp^2$, leading to a-C:Me coatings [15,18]. Nonmetals or small amounts of carbide formers do not show such a sharp decline in hardness reduction compared to metal ones [1]. Thus, doped coatings (ta-C:X) with properties similar to ta-C are formed, for instance, with regard to high $sp^3/sp^2$ ratios. In many cases, a significant stress reduction is observed due to doping with metals or nonmetals [19].

Important for the comparison of the effects of the doping elements is the erosion behavior of cathodes during arc evaporation. The erosion behavior of pure cathodes is well analyzed [8,9,20–29], but a lack of information about compound cathodes still exists.

Kimblin [25,26] and Daalder [21–23] show that the erosion rate $E_r$, representing a mass loss per electric charge ($\mu g/C$), includes three main influences (see Equation (1)):

$$E_r = E_{ion} + E_{MP} + E_{gas} \tag{1}$$

The erosion rate is caused by ion evaporation ($E_{ion}$), by macroparticles ($E_{MP}$) and by gas formation ($E_{gas}$), whereby the latter case plays a subordinate role.

The ion erosion rate $E_{ion}$ represents a lower limit for the erosion rate of a cathode material, and is only influenced by the arc current [23,30]. Daalder calculates this minimal erosion rate for pure carbon cathodes, where only ions contribute to the erosion with 13.16 $\mu g/C$ [21,23]. For graphite composite cathodes, no information on erosion rates are available. For understanding the erosion behavior, it is not only the cathode material that is important; the size and structure of the cathode are also considered to be relevant [21,25,26].

**Table 1.** Effect of dopants and selected references.

| Dopant | Intended Effect of the Dopant | Reference |
|---|---|---|
| Boron (B) | Reduction of intrinsic stress while maintaining high hardness | [19,31–35] |
| Silicon (Si) | Increase in thermal stability, surface smoothening | [36,37] |
| Iron (Fe) | Reduction of intrinsic stress, change of wetting properties | [15,38–40] |
| Molybdenum (Mo) | Reduction of intrinsic stress, reduction of electrical resistivity, improved wear behavior | [18,41] |

In this study, the effect of doping was comprehensively investigated by using sintered composite cathodes made from graphite and doping element powders. Two nonmetals (B, Si) and two metals (Fe, Mo) were investigated as doping elements with the aim to tailor the coating properties in particular directions, as summarized in Table 1.

In recently published works, it was shown that doping with some elements can also reduce the surface roughness significantly, which is likely caused by a low emission of particles during evaporation [17] or due to smaller particles [42].

## 2. Materials and Methods

The experimental setup is shown in Figure 2. Evaporation was carried out in a high vacuum at a pressure of about $10^{-4}$ Pa. A pulsed vacuum arc discharge of the graphite cathode (Figure 2, (1)) produced the plasma (Figure 2, (8)), which mainly consists of positively charged carbon ions, electrons and, in the case of compound targets, also the positively charged ions of the doping element. Furthermore, neutrals and macroparticles were formed during the evaporation process. The repetitive ignition of the arc pulses was triggered by a pulsed laser using a commercial Q-switched, Nd-doped yttrium aluminum garnet (Nd-YAG) laser (Figure 2, (2)). The laser pulse length was about 100 ns, with an energy of about 15 mJ, which is smaller than the arc discharge energy by a factor of 1000 (approx. 15 J). The combination of the linear scanning of the laser spot and the rotation of the graphite target results in uniform erosion. The employed current source provides a sinusoidal current of up to 1600 A at a discharge duration of 330 $\mu s$. A pulsed bias voltage of 100 V was applied synchronously to the arc pulses. The duration of the bias pulse was set to 175 $\mu s$.

Due to the reduced height of the modified graphite targets (180 mm in diameter at 60 mm height), and to allow low coating temperatures for these studies, the repetition rate was set to 50 Hz. The substrate ((7) in Figure 2) was set in a two-fold rotation and was equipped with a bias voltage source ((4) in Figure 2), which allowed defined bias pulse overlaps. A magnetron-sputtering source ((6) in Figure 2) was used to deposit a chromium interlayer to ensure the sufficient adhesion of the carbon coatings.

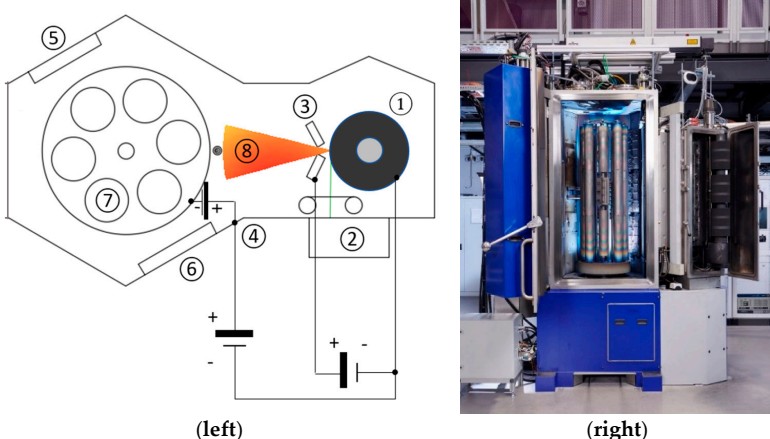

|                |                |
| :------------: | :------------: |
| **(left)**     | **(right)**    |

**Figure 2.** Scheme of the coating device DREVA 1200 (**left**): (1) cathode; (2) laser scanning system; (3) two-part anode; (4) bias supply; (5) radiation heater; (6) magnetron sputtering source; (7) substrate holder and rotation; (8) plasma; and coating chamber with Laser-Arc Module[TM] LAM500 (**right**). Photo by ©Jürgen Jeibmann, Dresden, Germany.

The main objective of this investigation was to study the evaporation process of modified and pure graphite cathodes and the properties of deposited doped and pure carbon coatings. For this purpose, pure graphite and sintered graphite compound cathodes with nominal amounts of 5 at.% B, Si, Fe or Mo were used to synthesize the ta-C and doped carbon coatings. In contrast to a standard cathode arrangement in the laser-arc process, with two graphite cylinders and a total cathode length of 40 cm, only small cylinder segments consisting of 3 discs each with a total height of 6 cm were used in this work. For the investigation of the chemical composition of the (t)a-C:X coatings as a function of height position in relation to the cathode position, several samples were mounted across the entire effective coating height (~60 cm) within the coating chamber (see Figure 3).

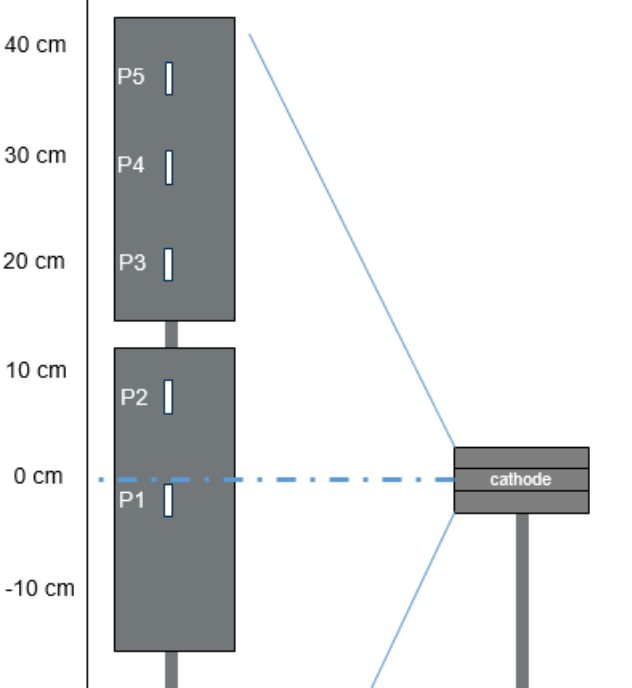

**Figure 3.** Scheme of the vertical sample position relative to center of the cathode (P1–P5) with a minimal distance of 65 cm (cathode surface (Figure 2; no. 3) to the sample holder (Figure 2; no. 7) and a maximal distance of 100 cm.

The coatings were deposited on flat steel samples (hardened low-alloy chromium steel (100Cr6, EN 1.3505, SAE 52100, Nosta GmbH, Höchstädt, Germany) with a size of 18 mm × 13 mm × 3 mm, polished to Ra < 20 nm). Prior to deposition, the samples were cleaned in an ultrasonic bath with an alkaline solution and then dried. For coating, they were fixed on a holder in an 8-axis planetary system and coated in a twofold rotation. The coating process started with an argon ion etching step using a hollow cathode. Next, a 100 nm thick Cr adhesion layer was deposited by magnetron sputtering, followed by (t)a-C:X deposition using an unfiltered laser-arc.

The evaporation process was characterized with respect to the lateral arc spot movement and erosion rates of the cathodes. The dimensions of the arc spot traces on the cathodes were measured after an evaporation of 1000 pulses. These values were averaged and put in relation to the respective evaporation current. By weighing (precision balance PCB 10000-1 Kern&Sohn GmbH, Balingen-Frommern, Germany) the cathodes prior and after the evaporation, the gravimetric erosion rate $E_{r,g}$ was calculated in relation to the electrical charge during the evaporation. The density of the cathodes was calculated from the initial weight and volume. With this information, the volumetric erosion rate $E_{r,v}$ could be derived. The deposition process and the prepared samples were characterized with respect to their deposition rate $D_r$ and chemical composition. The deposition rate was calculated according to the coating thickness and effective deposition time from the number of ignited pulses and the pulse frequency. By means of the ball crater-grinding method (KSG110 from Inovap Dresden, now HEF Group, Andrézieux-Bouthéon, France), the coating thickness was measured according to DIN EN ISO 26423. For the analysis of the chemical composition, all samples were measured using SEM (acceleration voltage of 10 V, spot size 50 and working distance of 10–13 mm) with an EDS-system JEOL 6610 + X-Max 80 mm$^2$ (JEOL, Akishima, Japan and X-MAX 80 from Oxford Instruments plc., Abingdon, United Kingdom). The chemical composition was determined using AZtec software (version: 3.3).

## 3. Results and Discussion

### 3.1. Coating Properties and Deposition Rate

A summary of selected coating properties from sample position P1 (Figure 3) is given in Table 2. Additional information about the coating properties can be found elsewhere [42]. All coatings were prepared with a similar coating thickness. Coatings containing nonmetal dopants have mechanical properties and deposition rates comparable with undoped ta-C, whereby a slight reduction with dopant amounts of around 5 at.% could be observed. In the case of metal dopants, hardness and Young's modulus and the deposition rates were drastically reduced, with a simultaneously higher doping content in the coating. The highest deposition rate is obtained for undoped ta-C, while the lowest deposition rate is observed for a-C:Fe, with the highest amount of dopant in the coating.

**Table 2.** Properties of ta-C and (t)a-C:X coatings, mounted on sample position P1; summary from [42].

| Target | Resulting Coating | Amount of Dopant (at.%) | Coating Thickness d (µm) | Deposition Rate $D_r$ (µm × 10$^{-6}$ Pulses) | Indentation Hardness $H_{IT}$ (GPa) | Young's Modulus $E_{IT}$ (GPa) |
|---|---|---|---|---|---|---|
| C (pure) | ta-C | - | 4.9 ± 0.4 | 3.3 | 52.5 ± 0.9 | 541 ± 15 |
| Nonmetal dopants | | | | | | |
| C-B | ta-C:B | 5.0 | 4.2 ± 0.3 | 2.5 | 50.9 ± 0.5 | 530 ± 9 |
| C-Si | ta-C:Si | 5.7 | 4.3 ± 0.4 | 2.4 | 44.8 ± 0.2 | 497 ± 5 |
| Metal dopants | | | | | | |
| C-Fe | a-C:Fe | 10.3 | 4.0 ± 0.4 | 0.7 | 14.4 ± 0.1 | 168 ± 4 |
| C-Mo | a-C:Mo | 7.4 | 4.9 ± 0.5 | 1.2 | 25.2 ± 0.3 | 323 ± 5 |

The elemental distribution within the doped coatings was analyzed by cross-section EDS analysis in SEM (Figure 4). Fe-containing coatings show a very slight increase in Fe from the interface to the surface of the coating, whereas the other coatings show no

significant gradient over the coating thickness. Therefore, no evidence for macroscopic variations in composition or temporal effects on deposition can be detected. The significant drop in the measured signal for ta-C:Si and a-C:Mo is an artefact of the metallographic preparation. Here, a narrow gap has formed between the sample and the embedding medium, which led to the observed drop.

For every sample position in relation to the cathode (see Figure 3), the chemical composition of the coatings was evaluated. The results of the position-dependent compositions for all doped coatings are shown in Figure 5. From the plasma investigations [1], it is known that the emission of species into the vacuum chamber shows an angular distribution. Due to this angular distribution, an elemental distribution of dopants in the deposited coatings may occur. Furthermore, the varying absolute and horizontal distance (see Figure 3) between the cathode and the position of each sample relative to the plane of the cathode could result in a variation in the amount of dopant in the coating. In the case of ta-C:B, the coating composition shows almost no deviation in the chemical composition in relation to the sample position, and the content of B corresponds to the nominal amount of B in the cathode material. In the case of Si, a slight deviation in the coating composition from the nominal Si amount in the cathode and in its dependence on the sample position is observed. Along the plane of the cathode's center, the amount of Si is around 6 at.%; with increasing distance, the amount of Si decreases down to 4 at.%.

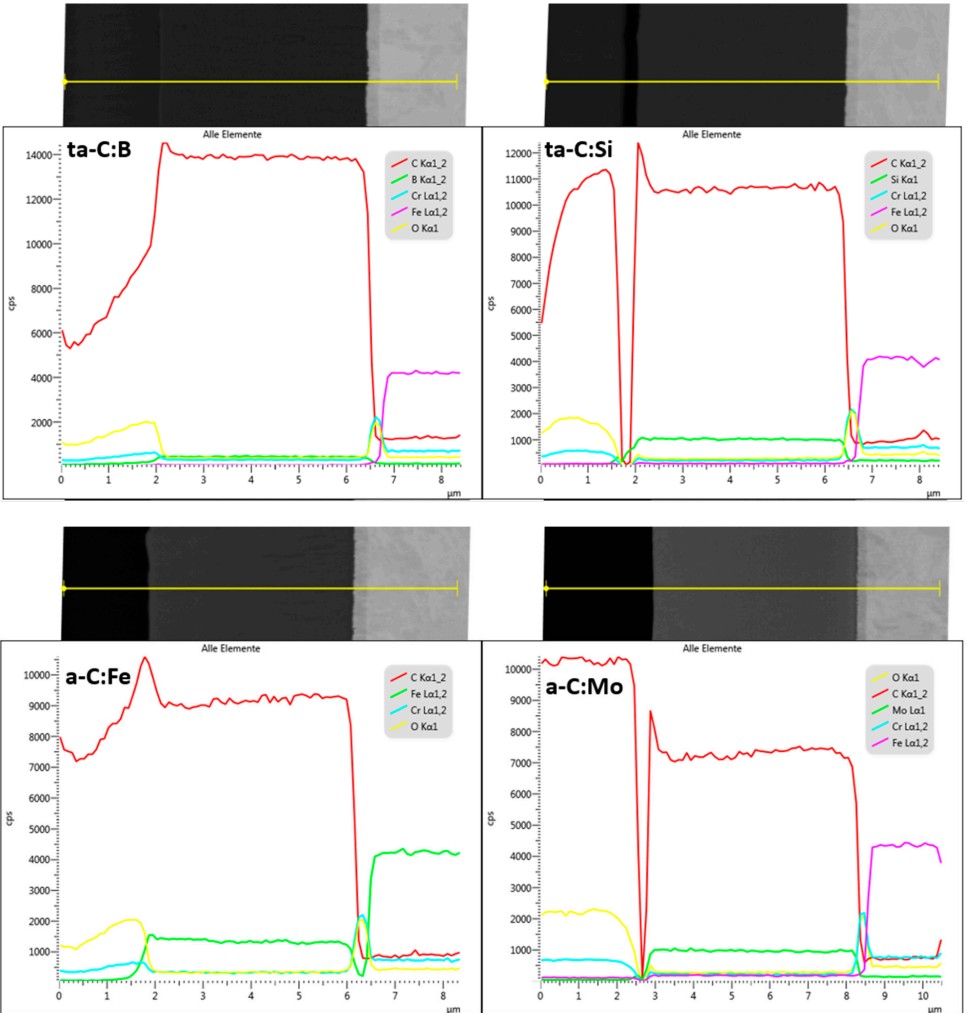

**Figure 4.** EDS line scan measurements with the signals of the identified elements (see colored lines) on cross-sections of doped carbon coatings. In the SEM pictures, the indexed position of the line scan is shown (yellow lines; scale bar of SEM pictures corresponds to the *X*-axis).

In the case of metals as doping elements, a more pronounced distribution in the chemical composition in relation to the sample position occurs, especially for Fe. On the sample along the plane at the cathode's center, the Fe content reaches a maximum of around 10 at.%, which is twice as high as the nominal amount of Fe in the cathode. Chen et al. [15] find a similar behavior of Fe enrichment in coatings caused by several effects. A preferential evaporation of Fe, due to the differences in the melting temperature of Fe and C, could be an explanation. By increasing the distance to the plane of the cathode center, a continuous decrease in Fe content down to around 5 at.% is clearly visible. A similar behavior is obtained by doping with Mo. The sample position along the plane at the cathode center leads to a slightly higher amount of Mo compared with the nominal cathode composition, but towards the outer positions, a decrease in the amount of Mo is recognizable. The reduction in hardness and Young's modulus in the metal-doped coatings (see Table 2 and [42]) is attributed to the effect on the formation of $sp^3$-bonded C-bonds. Thus, it can be assumed that the relatively heavy metal atoms hinder the formation of three-dimensional $sp^3$ structures and, thus, favor the formation of the (planar) $sp^2$ structures.

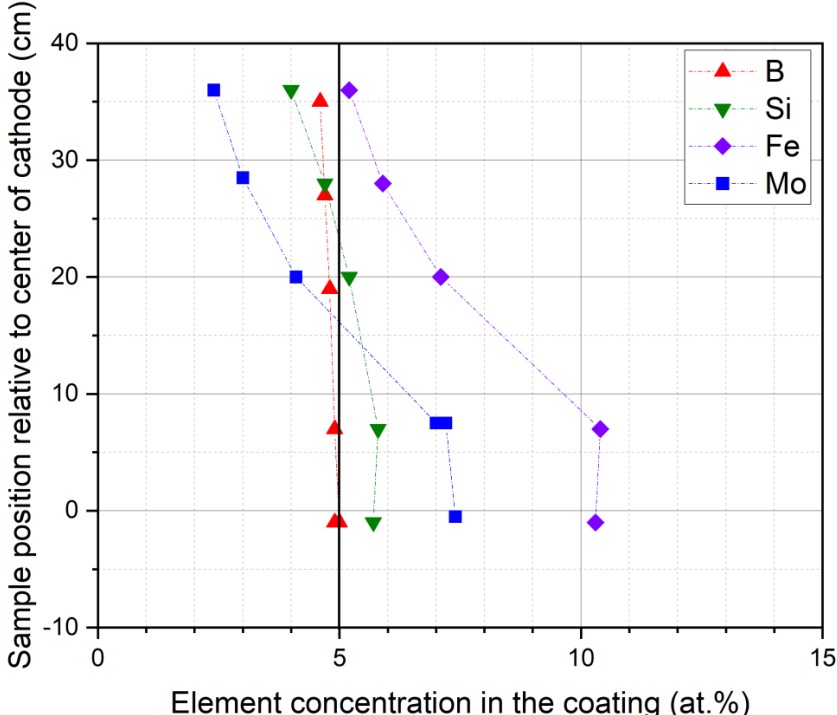

**Figure 5.** Element concentration over the vertical sample position relative to the center of the cathode; the nominal element concentration in the cathode is 5 at.%.

Finally, investigations were carried out on the deposition rate as a function of the vertical sample position. First, it should be clarified whether the deposition rate for the distribution of the doped coatings differs from pure ta-C. For comparison, the absolute and the normalized (to the highest value of each distribution) deposition rate is shown in Figure 6. In general, the deposition rate decreases towards the sample positions away from the plane of the cathode center. It should be mentioned that the plasma focus is slightly shifted upwards, also causing a shift in the deposition rate and the chemical content. Undoped ta-C has the highest absolute deposition rates for all the sample positions. B and Si doping leads to a significant reduction in the deposition rate by about 25%. However, a drastic reduction occurs in the case of Mo and Fe dopants: here, the deposition rate is reduced by about 65% (Mo) and about 80% (Fe) at the maximum.

Three possible reasons for the differences in deposition rates are: (a) the evaporation or erosion rates, (b) the possible shadowing of the plasma at the anode slot due to arc spots

of different sizes or (c) a different distribution of the plasma in the coating chamber. The latter reason obviously plays a subordinate role, as the normalized representation of the distributions (see inlay in Figure 6) shows a somewhat broader distribution only for the case of Fe doping.

In the next section, the influence of the erosion rate (a) and the trace of the arc spot (b) on the deposition rate will be investigated.

### 3.2. Cathode Erosion and Arc Spots

In order to understand the influence of doping elements on the deposition rates, investigations were carried out on the erosion of the cathodes during arc evaporation, as well as on the expansion of the arc spots on the cathode surface (Figure 7). For all cathodes, the lateral size of the traces of arc spots in relation to the evaporation current is shown in Figure 8. Generally, by increasing the arc evaporation current from 800 to 1600 A, the arc spots tend to spread in a more pronounced manner over the cathode surface, leading to an increase in the arc spot movement size. For B-modified and pure graphite cathodes, the arc spot movement is low, whereas the arc spots on Fe- and Si-modified graphite cathodes show a pronounced further spread of the arcs. The measurements may suggest that the expansion of the arc spots scales indirectly and proportionally to the cohesive energy of the cathode material. Values for the cohesive energy of the graphite and elements added to the graphite cathode are listed in Table 3. Additionally, an influence of the electrical conductivity or microstructure (e.g., porosity) of the cathode material on the movement of arc spot on the cathode surface was discussed for different types of graphite by Kandah et al. [6,8,9].

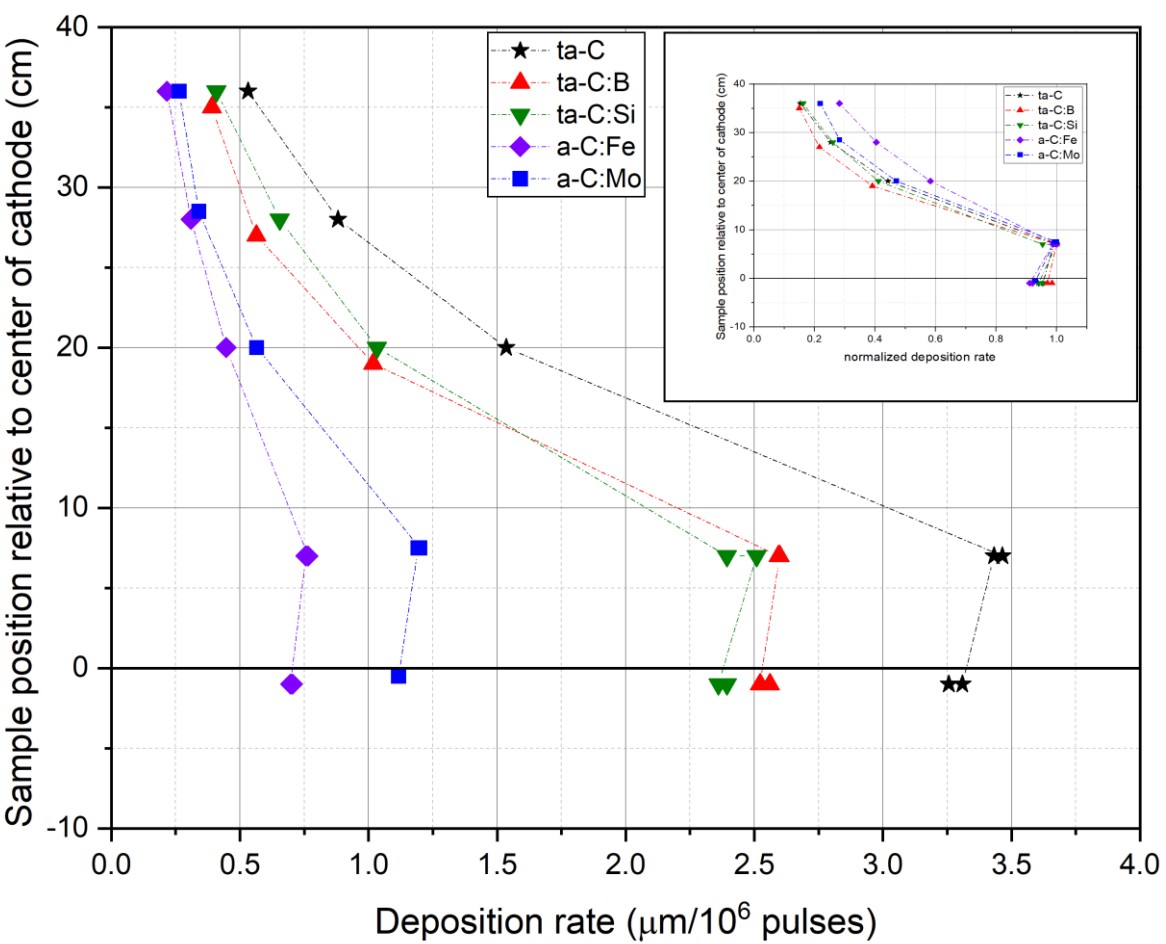

**Figure 6.** Absolute and normalized (inlay) deposition rates of different sample positions relative to the cathode center.

In Table 4, the calculated erosion rates of graphite and modified graphite cathodes are shown. For pure carbon cathodes, the highest erosion rate is observed. By modifying the graphite cathode, the erosion rate for all types is reduced. The reduction in the erosion rate for Mo- and Fe-modified cathodes is more pronounced than for B- and Si-modified ones.

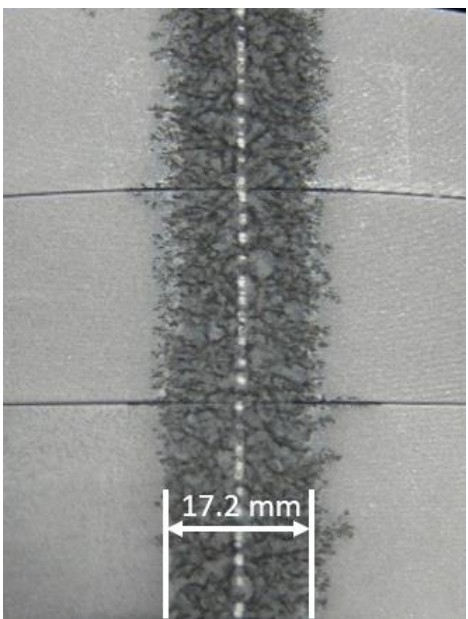

**Figure 7.** Example of traces of arc spots on graphite cathode surface with an arc current of 1600 A. The calculated lateral arc spot movement length is 17.2 mm.

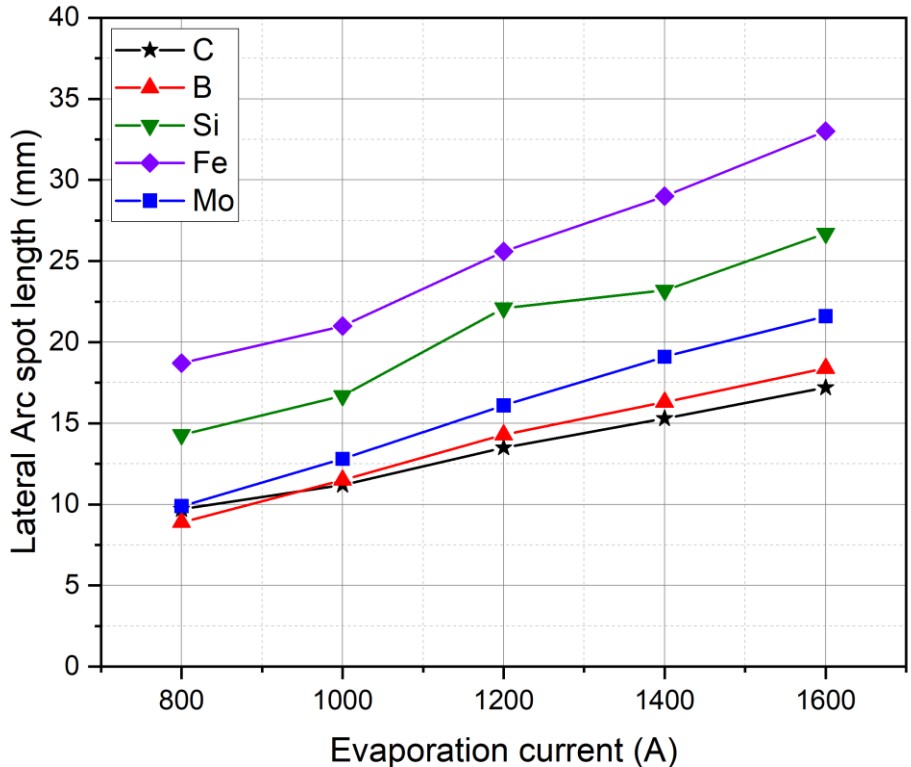

**Figure 8.** Lateral size of arc spot traces on the cathode surface in relation to the evaporation current.

**Table 3.** Atomic mass and cohesive energy of carbon and doping materials.

| Material | Atomic Mass | Cohesive Energy (eV/atom) |
|---|---|---|
| Iron (Fe) | 56 | 4.28 [1] |
| Silicon (Si) | 28 | 4.63 [1] |
| Molybdenum (Mo) | 96 | 6.82 [1] |
| Carbon (C) | 12 | 7.37 [1] |
| Boron (B) | 11 | ~8 [43] |

**Table 4.** Comparison of gravimetric and volumetric erosion rate and calculated density of target material for an evaporation current I = 1600 A.

| Target | Calculated Density of Target Material (g/cm$^3$) | Gravimetric Erosion Rate $E_{r,g}$ ($\mu$g/C) | Volumetric Erosion Rate $E_{r,v}$ ($10^{-5}$ cm$^3$/C) |
|---|---|---|---|
| C (pure) | 1.85 ± 0.01 | 47.7 ± 3.1 | 2.9 ± 0.3 |
| | | Nonmetal dopants | |
| C-B | 1.83 ± 0.01 | 34.1 ± 1.7 | 1.9 ± 0.1 |
| C-Si | 1.95 ± 0.06 | 36.5 ± 1.9 | 2.0 ± 0.2 |
| | | Metal dopants | |
| C-Fe | 2.44 ± 0.01 | 17.0 ± 4.1 | 0.7 ± 0.2 |
| C-Mo | 2.62 ± 0.02 | 27.2 ± 6.6 | 1.0 ± 0.2 |

As described above (Equation (1)), the erosion rate consists predominantly of an ion erosion rate $E_{ion}$ and a macroparticle erosion rate $E_{MP}$. The ion erosion rates known from the literature are about 13 $\mu$g/C for graphite [21,23], 40–50 $\mu$g/C for metallic Fe [23] and 50–55 $\mu$g/C for metallic Mo [23] (no data are available for pure B and Si because these materials are not evaporable in the arc process). Surprisingly, adding Fe or Mo to the graphite does not increase the erosion rate, but decreases it drastically, as can be seen in Table 4. This cannot have anything to do with the macroparticle emission rate $E_{MP}$, because the Fe- and Mo-doped coatings tend to show even fewer particle-induced defects than in the case of pure ta-C [42]. The drastic reduction in the erosion rate of composite cathodes compared to pure graphite must therefore have other reasons. We suspect that this is related to the local electrical conductivity in the composite graphite, which influences the formation of current paths in the cathode. Even in pure graphite, Kandah [6] has found strong differences in erosion rates depending on the graphite type, i.e., the structure of the graphite cathode.

A higher erosion rate $E_{r,v}$ results in a higher deposition rate $D_r$. This almost linear relationship is illustrated in Figure 9. A pure carbon cathode with the highest erosion rate of around $E_{r,v} = 2.9 \times 10^{-5}$ cm$^3$/C exhibits also the highest deposition rate $D_r = 1.3$ $\mu$m/C compared to the modified cathodes. Corresponding to the reduction of the erosion rate, the deposition rate is also moderately reduced in the case of nonmetals (B, Si), and drastically so in the case of metals (Fe and Mo).

The linear relationship makes it clear that the erosion rate is the primary cause of the differences in deposition rates. All other influences, such as the shading of the plasma as it passes through the anode shield or different plasma distributions in the chamber, are secondary in their effects.

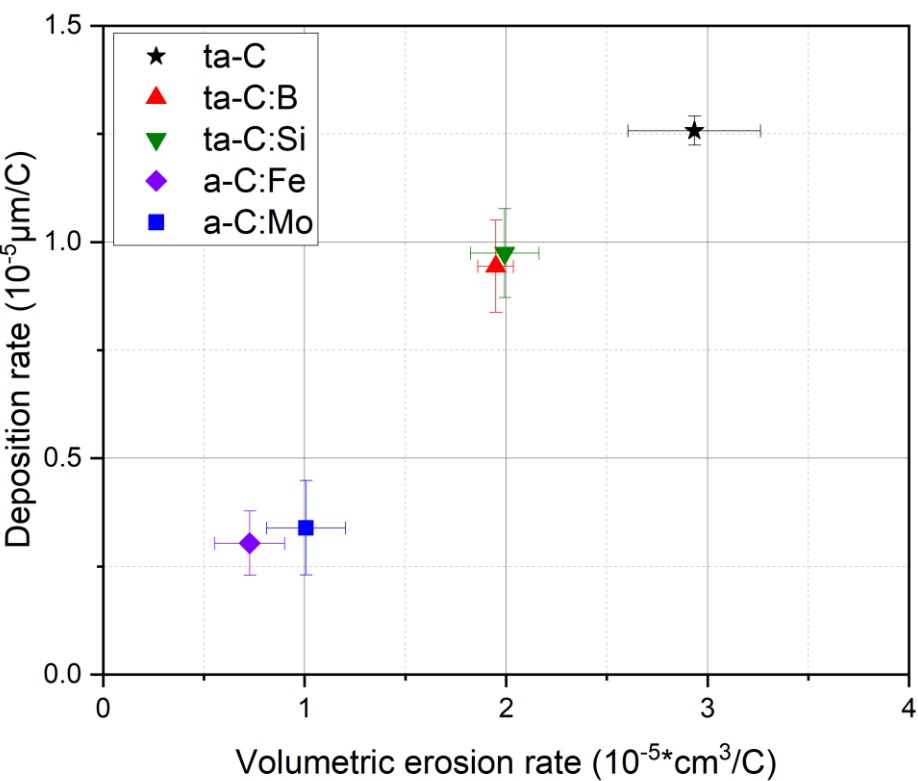

**Figure 9.** Deposition rate $D_r$ over the volumetric erosion rate $E_{r,v}$ of coatings for sample position P1; mounted along the plane of the cathode center.

## 4. Conclusions

In this paper, pure ta-C and doped carbon coatings with metal and nonmetal elements were deposited using a laser-arc evaporation process. Both the influence of the added elements on the cathode erosion and the influence on the deposited (t)a-C:X coating properties were investigated. It was shown that doped (t)a-C:X coatings with homogeneous element distribution can be reliably produced with the laser-arc process. By analyzing the homogeneity by means of EDS measurements, no evidence of chemical gradient or temporal effects of the chemical deposition was observed. Investigations into the deposition rate and the chemical composition in relation to the vertical sample position revealed a similar deposition behavior with respect to the deposition rate distribution. Slight deviations in the chemical content of the coatings, differing with the sample position, were found for nonmetals, whereas metal doping led to a strong deviation in the chemical content of the coating. The deposition rates for the doped coatings are lower than for pure ta-C. A drastic reduction was observed in the case of the metal dopants (Fe and Mo). It was shown that the greatest influence of the deposition rate lies in the erosion rate during arc discharge on the cathode. However, minor influencing factors can also be the height distribution of the plasma as well as the lateral expansion of the arc spots, in combination with the shadowing effect of the anode shield.

Further investigations should be conducted using TEM and Raman analysis to clarify the coating structure, how dopants are incorporated in the carbon coatings and how they change the properties of coatings in detail. Furthermore, the extent to which the doped coatings can be deposited on parts with more complex geometry, and whether there are geometric influencing factors, should also be investigated. A final important object of investigation would be to find the cause of the strong differences in erosion rates for the different dopants in the cathodes, and the extent to which this has something to do with the emission of macroparticles.

**Author Contributions:** Conceptualization, T.K., F.K. (Florian Kirsten) and V.W.; methodology, T.K. and F.K. (Frank Kaulfuß); validation, T.K., F.K. (Florian Kirsten) and F.K. (Frank Kaulfuß); formal analysis, T.K. and F.K. (Florian Kirsten); investigation, F.H. and F.K. (Florian Kirsten); resources, F.H.; data curation, T.K., F.K. (Frank Kaulfuß) and F.K. (Florian Kirsten); writing—original draft preparation, T.K.; writing—review and editing, T.K., F.K. (Frank Kaulfuß) and V.W.; supervision, V.W.; project administration, F.K. (Frank Kaulfuß) and V.W. All authors have read and agreed to the published version of the manuscript.

**Funding:** This research was funded by the German Federal Ministry of Economic Affairs and Energy (BMWi), grant numbers 03ET1609 E.

**Institutional Review Board Statement:** Not applicable.

**Informed Consent Statement:** Not applicable.

**Data Availability Statement:** Data available on request due to restrictions/data sharing not applicable.

**Acknowledgments:** The authors would like to thank the colleagues and students of the Department of Carbon Coatings at Fraunhofer IWS and Jörg Kaspar and Martin Kuczyk from the Group of Materials and Failure Analysis.

**Conflicts of Interest:** The authors declare no conflict of interest. The funders had no role in the design of the study; in the collection, analyses, or interpretation of data; in the writing of the manuscript, or in the decision to publish the results.

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
