# Peer review of "Amorphous Carbon Coatings with Different Metal and Nonmetal Dopants: Influence of Cathode Modification on Laser-Arc Evaporation and Film Deposition"

_coatings, doi:10.3390/coatings12020188_

Round 1
Reviewer 1 Report
In this paper, the authors fabricated amorphous carbon coatings with different metal and non-metal dopants and demonstrated the influence of cathode modification on laser arc evaporation and film deposition. The paper is well presented, and data was solid. And thus, I recommend the publication of this paper in Coatings after minor revision. Some questions and suggestions are listed as follows:
1. The SEM images of arc spot movement length may also be included in the article for further evidence of the expansion in doped graphene cathodes.
2. Some experiments may be added to better verify the possibility of drastic reduction in the deposition rates of Fe and Mo.
3. When exploring the interpretation of the corrosion rate, it is demonstrated in this paper that the corrosion rate is related to the conductivity. Can it supplement the trend of conductivity in the doped graphene cathode?
Author Response
We thank the reviewer for the positive assessment of our manuscript. Our responses to the reviewer's comments are as follows:
- The SEM images of arc spot movement length may also be included in the article for further evidence of the expansion in doped graphene cathodes.
- Until now SEM analysis of the cathodes could not be implemented, due to the fact, that we still use the cathodes for evaporation purposes. We agree, more detailed analyses on the arc traces of the doped cathodes are quite reasonable. In a further study, we plan to carry out more detailed SEM investigations on this.
- Some experiments may be added to better verify the possibility of drastic reduction in the deposition rates of Fe and Mo.
- This is indeed one of the most important questions raised by this work. We plan to pursue this question in a follow-up paper. For this purpose, we want to perform more detailed analyses on the target material as well as investigate the question of what is the direct cause for the strong differences in the erosion rates by means of process variations.
- When exploring the interpretation of the corrosion rate, it is demonstrated in this paper that the corrosion rate is related to the conductivity. Can it supplement the trend of conductivity in the doped graphene cathode?
- Until now, we only have inconsistent conductivity data (obtained on the cathodes themselves), so that we need precise measurments on small samples from the cathode – which could be implemented after we will no longer use the cathodes for evaporation purposes (see ques. No. 1).
Reviewer 2 Report
Coatings-1567230
Amorphous Carbon Coatings with Different Metal and Non-metal Dopants: Influence of Cathode Modification on Laser-Arc Evaporation and Film Deposition
Amorphous carbon coatings with different metal and non-metal dopants were prepared.
In this work pure ta-C and doped carbon coatings with metal and nonmetal elements were deposited using a Laser Arc evaporation process.
The effect of doping was comprehensively investigated by using sintered composite cathodes, made from graphite and doping element powders. Two nonmetals (B, Si) and two metals (Fe, Mo) were investigated as doping elements to tailor the coating properties in particular directions.
Results showed a precise understanding of the carbon coatings with different metals and non-metals. Well written and well presented with adequate figures, tables, references.
The article can be accepted in its present form.
With Regards,
Author Response
We thank the reviewer very much for the positive assessment of our manuscript.
Reviewer 3 Report
Journal: Coatings
Manuscript ID: coatings-1567230
This manuscript presents the results on “Amorphous Carbon Coatings with Different Metal and Non-metal Dopants: Influence of Cathode Modification on Laser- 3 Arc Evaporation and Film Deposition” by Tim Krülle et.al.
The manuscript looks good and is very well written. However, before the final decision, I kindly request the authors to answer the following questions:
- Authors should elaborate the synthesis process of modified graphite targets (eg: sintering temperature, composite mixing) mentioned in (line number 130-134)
- In figure 3, position P6 is missing.
- Usually, the sophisticated techniques are known for their precession to coat in nanometric level homogeneously. Can authors comment on the considerable deviation (>15 %) in the coating thickness of all the samples?
- After restricting the coating thickness to the nanometre level, is there any significant difference in the materials property? (Authors can refer to this article https://doi.org/10.1016/j.jeurceramsoc.2012.06.026)
- The indentation hardness of the C-Fe target is ~260 % lower than that of the (C-pure) sample. Could you please comment on this? If the formation of the sp2 carbon is the main reason for this, could you please illustrate and possibly quantity the formation of sp2 and sp3 using Raman spectroscopy analysis? (Authors may refer to this following article https://doi.org/10.1016/j.carbon.2018.11.038) otherwise, necessary referencing is needed.
- The authors should provide the scale for the SEM images shown in Figure 4. Also, according to the reported literature, it will be better to see the homogeneity of the coating with the top view images and elemental mapping.
- The authors should explain the anomaly observed for the carbon signals for the Mo (~ 2.6 µm) and Si (~1.7 µm) doped carbon samples in Figure 4.
- Please mention the current erosion rate in Table 4?
Minor comments
- In Table 2, please change the (,) to (.) for the indentation hardness of C-Fe (14.4 ± 0,1 to 14.4 ± 0.1.)
- Figure 6 X-axis label.
Author Response
We thank the reviewer for the positive assessment of our manuscript. We have further improved the manuscript according to the reviewer’s comments and suggestions:
- Authors should elaborate the synthesis process of modified graphite targets (eg: sintering temperature, composite mixing) mentioned in (line number 130-134)
- The cathodes were supplied by a commercial manufacturer. The (only) specification was: 5at% foreign element concentration, 95 at% graphite. Unfortunately, the manufacturer does not disclose any information about the manufacturing details for technology protection reasons.
- In figure 3, position P6 is missing.
- Thanks for the hint. The range is only until P5; it has been corrected in the manuscript.
- Usually, the sophisticated techniques are known for their precession to coat in nanometric level homogeneously. Can authors comment on the considerable deviation (>15 %) in the coating thickness of all the samples?
- You are right the coating thickness homogeneity is generally very good with PVD coatings, also the reproducibility should be close to 100% - under the same boundary conditions. However, for different cathode materials, as we used in our investigation, the predictability of the coating thickness is very limited. Although we carried out a preliminary test in each case to hit the targeted film thickness, it would have taken considerably more effort to achieve an exact film thickness in all cases. For the investigations in this paper, we have accepted the film thickness variations of up to 15% without any doubts about the basic conclusions.
- After restricting the coating thickness to the nanometre level, is there any significant difference in the materials property? (Authors can refer to this article https://doi.org/10.1016/j.jeurceramsoc.2012.06.026)
- We have focused on coating thicknesses (4-5 µm) that are within the range of common tribological coating applications for components and tools. At this thickness, the coatings are to be considered homogeneous.. At the nanometer level, however, there could be some degree of heterogeneity due to specific doping-element induced nanostructure. On this scale, we then actually expect significantly different material properties than with thick coatings. By the way, we are currently working on a paper that deals with the structural investigation of these coatings in the nanometer range and which also addresses the fracture mechanical properties.
- The indentation hardness of the C-Fe target is ~260 % lower than that of the (C-pure) sample. Could you please comment on this? If the formation of the sp2 carbon is the main reason for this, could you please illustrate and possibly quantity the formation of sp2 and sp3 using Raman spectroscopy analysis? (Authors may refer to this following article https://doi.org/10.1016/j.carbon.2018.11.038 ) otherwise, necessary referencing is needed.
- We suspect that behind the hardness reduction there is mainly a strong reduction of the sp3 content. A further work focusses on the Raman and TEM investigation, but for this paperwe focussed on the erosion and deposition behaviour of the coatings. The Properties of the coatings are indeed very important, but should adress a following paper with the detailed structural investigations.
- The authors should provide the scale for the SEM images shown in Figure 4. Also, according to the reported literature, it will be better to see the homogeneity of the coating with the top view images and elemental mapping.
- We found no evidence of heterogeneous distribution of doping elements in any of the SEM studies. Therefore, the top-view views and a top-view element mapping do not add any value.We focussed on studying temporal effects during the deposition process and, hence, EDS measurement on the polished cross sections.
- The authors should explain the anomaly observed for the carbon signals for the Mo (~ 2.6 µm) and Si (~1.7 µm) doped carbon samples in Figure 4.
- This anomaly is the effect of the metallopgraphic sample preparation; a small gap between sample and the embedding material could occur, which lead to a decrease of the signal.
- Please mention the current erosion rate in Table 4?
- The current is I=1600 A. It has been added to the manuscript.
Minor comments
- In Table 2, please change the (,) to (.) for the indentation hardness of C-Fe (14.4 ± 0,1 to 14.4 ± 0.1.): You are right. It has been corrected.
- Figure 6 X-axis label: The X-axis label is already shown in the figure below the X-axis or is there another suggestion for the X-axis label?
Reviewer 4 Report
The manuscript describes interesting and important results. The wording is clear, the data looks good, the conclusions are well supported. I recommend the manuscript for publication without modification.
The manuscript describes the production of doped amorphous carbon coatings and focuses on the influence of the experimental parameters used during the film deposition. Since the properties and applicability of amorphous carbon coatings are often hindered by lack of accurate control of the growth, this detailed investigation is an important contribution for the scientific community. There are several similar publications, but the authors achieved better control than most groups working in this field.
The manuscript is well written, the text is clear, the methods are described in details, the conclusions are well supported. I expected further investigations using TEM and Raman analysis to determine the quality of the coatings, but due to the length of this manuscript the authors plan to publish this in their next paper.
I recommend the publication of the manuscript.
Author Response

(The authors gave the same response as above.)

Round 2
Reviewer 3 Report
The authors have answered all the raised questions. I am happy with the answers. I propose the consideration of the manuscript in the present form.